# An Efficacious Transgenic Strategy for Triple Knockout of Xeno-Reactive Antigen Genes GGTA1, CMAH, and B4GALNT2 from Jeju Native Pigs

**DOI:** 10.3390/vaccines10091503

**Published:** 2022-09-08

**Authors:** Seungwon Yoon, Seulgi Lee, Chungyu Park, Hyunyong Choi, Minwoo Yoo, Sang Chul Lee, Cheol-Ho Hyun, Nameun Kim, Taeyoung Kang, Eugene Son, Mrinmoy Ghosh, Young-Ok Son, Chang-Gi Hur

**Affiliations:** 1Cronex Inc., Jeju 63078, Korea; 2Interdisciplinary Graduate Program in Advanced Convergence Technology and Science, Jeju National University, Jeju-si 63243, Korea; 3Cronex Inc., Cheongju 28174, Korea; 4Cronex Inc., Hwaseong 18525, Korea; 5Jeju Special Self-Governing Province Livestock Promotion Agency, Jeju 63078, Korea; 6College of Veterinary Medicine and Veterinary Medical Research Institute, Jeju National University, Jeju-si 63243, Korea; 7Department of Animal Biotechnology, Faculty of Biotechnology, College of Applied Life Sciences, Jeju National University, Jeju-si 63243, Korea; 8Department of Biotechnology, School of Bio, Chemical and Processing Engineering (SBCE), Kalasalin-Gam Academy of Research and Educational, Krishnankoil 626126, India

**Keywords:** CRISPR-CAS9 system, xenotransplantation, immune rejection, Jeju native pigs

## Abstract

Pigs are promising donors of biological materials for xenotransplantation; however, cell surface carbohydrate antigens, including galactose-alpha-1,3-galactose (α-Gal), N-glycolylneuraminic acid (Neu5Gc), and Sd blood group antigens, play a significant role in porcine xenograft rejection. Inactivating swine endogenous genes, including *GGTA1*, *CMAH*, and *B4GALNT2*, decreases the binding ratio of human IgG/IgM in peripheral blood mononuclear cells and erythrocytes and impedes the effectiveness of α-Gal, Neu5Gc, and Sd, thereby successfully preventing hyperacute rejection. Therefore, in this study, an effective transgenic system was developed to target *GGTA1*, *CMAH*, and *B4GALNT2* using CRISPR-CAS9 and develop triple-knockout pigs. The findings revealed that all three antigens (α-Gal, Neu5Gc, and Sd) were not expressed in the heart, lungs, or liver of the triple-knockout Jeju Native Pigs (JNPs), and poor expression of α-Gal and Neu5G was confirmed in the kidneys. Compared with the kidney, heart, and lung tissues from wild-type JNPs, those from *GGTA1*/*CMAH*/ *B4GALNT2* knockout-recipient JNPs exhibited reduced human IgM and IgG binding and expression of each immunological rejection component. Hence, reducing the expression of swine xenogeneic antigens identifiable by human immunoglobulins can lessen the immunological rejection against xenotransplantation. The findings support the possibility of employing knockout JNP organs for xenogeneic transplantation to minimize or completely eradicate rejection using multiple gene-editing methods.

## 1. Introduction

In contrast to most mammalian species, pigs are promising donors of biological materials for xenotransplantation because of their genetic, anatomical, and physiological similarities to humans [1,2,3,4,5,6,7,8,9,10]. However, despite these similarities, pigs and humans are phylogenetically distinct, which can lead to immune complications after xenotransplantation. Immune rejection is the most important obstacle to overcome in xenotransplantation. To suppress the adaptive immune response, exogenous immunosuppressive intervention is obligatory. However, the optimal immunosuppressive regimen needed after xenotransplantation is unknown, and whether this optimal regimen will be more intense than that required after allotransplantation is uncertain.

Thus, the prevention of delayed immune responses, particularly those targeting the endothelial lining of the graft vasculature, has received increasing attention. The expression of MHC class I, II, and III genes is closely correlated with T-cell-mediated immune responses, which represent a subset of adaptive immune responses. The task of presenting antigens on the cell surface for detection by CD8+ and CD4+ T-cells is performed by MHC class I and II proteins, respectively [11,12]. After xenotransplantation, porcine MHC molecules such as MHC class I and II proteins, also known as swine leukocyte antigen class I (SLA-I) and II (SLA-II) proteins, interact with anti-human leukocyte antigen antibodies and cause human T-cell responses [13,14]. Therefore, in pig-to-human xenotransplantation, SLA molecules are crucial to the cellular immune response. The knockout or knockdown of genes encoding SLA molecules may ensure the long-term survival of xenografts. Studies have generated donor pigs carrying SLA-I deficiency and demonstrated that genetically modifying *GGTA1*, *CMAH*, and *B4GALNT2* can effectively improve the survival of xenografts [15,16,17,18].

Important surface antigens responsible for immune rejection in human recipients include the N-glycolylneuraminic acid (Neu5Gc) and Sd(a) blood group antigens, both of which can cause acute vascular rejection given the human immune status and immunological challenges, as well as galactose-alpha-1,3-galactose (α-Gal), which causes hyperacute rejection [5,7]. The inactivation of the porcine endogenous genes such as *GGTA1*, *CMAH*, and *B4GALNT2* has been suggested to reduce the binding ratio of human IgG/IgM in peripheral blood mononuclear cells (PBMCs) and erythrocytes and inhibit the efficiency of α-Gal, Neu5Gc, and Sd(a), thereby effectively preventing hyperacute alleviating rejection [15,16,17,18] and improving the immune compatibility between humans and pigs to prolong graft survival [19,20]. The U.S. Food and Drug Administration recently approved the potential use of α-Gal-free pigs as a biomedical source.

The Jeju native pig (JNP) is found on Jeju Island, located in the Korean peninsula, and is a representative of the Korean native black pig. JNPs exhibit unique genetic characteristics and strong disease tolerance [21] and could be suitably engineered for use as clinical models. However, the biological basis for xenotransplantation using JNPs has not been clearly demonstrated.

In this study, we targeted the porcine genes *GGTA1*, *CMAH*, and *B4GALNT2* using the CRISPR-Cas9 system and introduced α-Gal, Neu5Gc, and Sd deficiency by somatic cell nuclear transfer (SCNT) to reduce antibody-mediated xenograft rejection after xenotransplantation. We investigate the immune rejection response of triple-knockout pigs in the kidneys, heart, lungs, and liver and validate their suitability before transplanting the heart and kidney into non-human primates.

## 2. Materials and Methods

### 2.1. Ethical Approval

The protocol for this study was approved by the Institutional Animal Care and Use Committees of Cronex Inc., Jeju, Korea (CRONEX-IACUC: 202002-003, CRONEX-IACUC: 202101-008, CRONEX-IACUC: 202101-009). The experiments were performed in compliance with relevant laws and regulations and the ARRIVE guidelines (PLoS Bio 8(6), e1000412,2010) [22]. Criteria for anesthesia and euthanasia were set according to the guidelines for laboratory animals of the Korean Ministry of Food and Drug Safety and the guidelines for animal euthanasia of the American Veterinary Medical Association.

### 2.2. Animals and Chemicals

JNPs were housed in the animal facility at Cronex Inc. (Jeju Special Self-Governing Province, South Korea) in a pathogen-free environment. The room was maintained at 24 ± 2 °C with 12 h light/12 h dark cycles. Filtered air and water as well as sterilized feed were supplied. All the pigs in this study had blood type O. To conduct additional experiments, 16-day-old healthy JNPs were humanely euthanized, and their organs were harvested. Under general anesthesia, the animals were euthanized by intravenous injection of 2 mmol/kg potassium chloride solution, and the death was certified by a veterinarian. Unless otherwise specified, all chemicals were purchased from Sigma-Aldrich (St. Louis, MO, USA).

### 2.3. Primary Cell Culture

The ear tissue of a 16-day-old JNP was washed with ethanol and Dulbecco’s phosphate-buffered saline (DPBS), and the hair and cartilage were removed. After chopping the ear skin, primary cells were separated by treatment with 0.25% trypsin. The primary cells were cultured in Dulbecco’s modified Eagle’s medium (DMEM) supplemented with 15% fetal bovine serum (FBS), 400 unit/mL penicillin, and 400 µg/mL streptomycin at 38 °C in 5% CO_2_ and 95% air. The medium was replaced daily. A large number of primary cells was secured through passaging.

### 2.4. Designing of Guide RNA (gRNA)

gRNA was designed using the CRISPR Design System developed by Benchling (San Francisco, CA, USA). Three types of gRNAs were designed for the *GGTA1*, *CMAH*, and *B4GALNT2* gene sequences obtained from the National Center for Biotechnology Information (NCBI) database (accession numbers NC_010443, NC_010449, and NC_010454, respectively). Each gRNA contained a specific restriction site (BsrI, PfoI, and DraⅢ) that disappeared after DNA double-strand break and subsequent non-homologous end joining by nucleotide insertion or deletion.

### 2.5. Preparation of Vector Expressing Cas9 and gRNA Targeting Triple Gene

pSpCas9(BB)-2A-Puro (Addgene Plasmid ID: 48139) was used as the source of the Cas9-triple fusion cassette. The vector was constructed as previously described [23] by restriction with Bbs I and insertion of oligo DNA transcribing gRNA targeting each gene. Three different vectors with different gRNAs targeting different sites of the exons, *GGTA1*-exon7, *CMAH*-exon4, and *B4GALNT2*-exon4, were constructed. After cloning the Cas9-triple fusion cassette, a Plasmid Maxi kit (Qiagen, Hilden, Germany) was used to obtain a high concentration of the vector for transfection.

### 2.6. Transfection of Primary Cell

Porcine ear skin cells were maintained in DMEM supplemented with 10% FBS, 100 unit/mL penicillin, and 100 unit/mL streptomycin at 38 °C in 5% CO_2_ and 95% air. Nucleofection was performed using the Neon Transfection System (Thermo Fisher Scientific) according to the manufacturer’s protocol. Briefly, 5 × 10^5^ fibroblasts were transfected with 5.7 μg plasmid DNA linearized by Not I. A day after transfection, cells were selected with 8 μg/mL puromycin for 5 days. Selected cells were sub-cultured for an additional 10 days, and emerging colonies were picked and subjected to genotyping using PCR and subsequent restriction fragment length polymorphism (RFLP).

### 2.7. Oocyte Collection and In Vitro Maturation

Ovaries were collected from the slaughterhouse immediately after slaughter and transported in a thermos. They were washed with saline solution at 38 °C upon arrival at the laboratory. A vacuum pump and an 18-gauge needle were used to aspirate cumulus-oocyte complexes (COCs). A follicle measuring more than 3 mm was chosen as the suction target. The aspirate was placed in a 50 mL tube and allowed to settle for 15 min before washing with T-HEPES (media 199, Sigma-Aldrich; 1% PVA, Sigma-Aldrich; sodium pyruvate, Sigma-Aldrich; 200 µM D-glucose; 3mM penicillin/streptomycin, Gibco 1835954; 1% sodium bicarbonate, s8875, Sigma-Aldrich). The sediment and T-HEPES mixture were poured into a 60 mm petri dish to locate COCs using a microscope. The collected COCs were washed three times in in vitro maturation media (Media 199, Gibco; 10% porcine follicular fluid, EGF 10 ng/mL, HCG 10 IU/mL, cAMP 100 μM, and PMSG 10 IU/mL) and cultured in a four-well dish. COCs were allowed to mature for 42 h, which consisted of culturing them for 22 h in hormone-supplemented media and for 20 h in hormone-free media.

### 2.8. Somatic Cell Nuclear Transfer and Embryo Transfer

Mature COCs from JNPs ovaries were pipetted to DPBS solution containing 0.1% hyaluronidase (H3506). MII oocytes were discovered in sucrose media after being washed three times in T-HEPES medium. They were removed from the enucleation media (T- HEPES with cytochalasin, Sigma-Aldrich C6762 7.5 g/mL) with a microinjection pipette to remove the polar body and MII chromosome. The donor cell was inserted after approximately an hour of rest. After being injected with donor cells, the oocytes were transferred to a 1 mm fusion chamber containing a 0.28 M mannitol solution. A cell manipulator (ECFG21; Nepa Gene, Chiba, Japan) was used to apply 130 V and 50 ms^−2^ pulse to activate electrolysis. The reconstructed oocytes were cultured in porcine zygote medium-5 at 39 °C and 5% CO_2_ after fusion and activation. The fusion was examined the next day. The reconstructed embryos were implanted into recipient JNPs. Approximately 170 reconstructed embryos were surgically transferred after being cultivated for 1 to 2 days. The fallopian tube was implanted with embryos, the wound was sutured, and, 30 days following the transplant, an ultrasound machine was used to check for pregnancy. Regular checks were conducted throughout the pregnancy, and the offspring was delivered naturally.

### 2.9. Extraction of DNA and Sequence Analysis

Genomic DNA was extracted from the harvested fibroblast samples using DirExTM Fast-Hair and Tissue and Tissue Plus! SV mini kits (GeneAll Biotechnology, Seoul, Republic of Korea) were used to extract genomic DNA from the umbilical cord of the transgenic pig. Extractions were performed according to the manufacturer’s instructions and subsequently used for PCR-RFLP. The CRISPR target region on each of the genes was amplified using PCR with Power S Taq DNA Polymerase Premix (HKGamp, Daejeon, Republic of Korea). Amplified DNA was restricted using Bsr I, Pfo I, or DraⅢ (for *GGTA1*, *CMAH*, and *B4GALNT2*, respectively) and electrophoresed using 2% agarose gel stained with RedSafe^TM^ (iNtRON, Seongnam-si, Gyeonggi-do, Republic of Korea). Sequencing was performed by Macrogen (Seoul, Republic of Korea) after TA cloning to confirm the correct DNA sequence of the triple-knockout cell lines verified using PCR-RFLP. Sequencing results were analyzed using NCBI’s BLAST global alignment, and translation analysis was performed using the Sahlgrenska Academy Program of the University of Gothenburg (http://bio.lundberg.gu.se/edu/translat.html; accessed on 1 September 2019).

### 2.10. Histologic Analysis and Immunofluorescence Microscopy

Tissue samples were fixed in 4% paraformaldehyde, dehydrated, embedded in paraffin, sectioned into 4 μm thick sections, and stained with hematoxylin and eosin (H&E). An Olympus BX51 microscope (Olympus, Tokyo, Japan) equipped with a DP71 digital camera was used for histological examination (Olympus).

To confirm the expression of α-Gal and Sd(a) antigens in porcine tissues, sections were prepared after dewaxing with xylene, rehydrating with gradient alcohol, and antigen unmasking with citrate solution. The PBS-washed slides were incubated with diluted Isolectin GS-IB4 (1:1000; Invitrogen, Waltham, MA, USA) or *Dolichos biflorus* agglutinin (1:400; Vector Laboratories) in the dark at 20–22 °C for 60 min. For confirmation of Neu5Gc expression in the tissues, a chicken anti-Neu5Gc antibody kit (BioLegend, San Diego, CA, USA) with Goat anti-Chicken IgY Alexa Fluor 488 (Invitrogen) as the secondary antibody was used to stain the antigen-unmasking slides. After washing with PBS, nuclear staining was performed using 4,6-diamidino-2-phenylindole (DAPI; Invitrogen) in all cases. To confirm binding to human antibodies, the slides were incubated with diluted heat-inactivated human serum for 30 min (diluted to 20% for IgM and 5% for IgG binding). After washing with PBS, the slides were blocked with 10% goat serum for 30 min at room temperature. To detect IgM or IgG binding, Goat anti-Human IgG Alexa Fluor 488 or Alexa Fluor 488-Donkey anti-Human IgM (1:1000; Invitrogen) was applied for 30 min in the dark at room temperature and DAPI was used for nuclear staining. Slides were examined under a fluorescence stereo microscope (M165FC, Leica Microsystems AG, Heerbrugg, Switzerland) at the Bio-Health Materials Core-Facility (Jeju National University).

## 3. Results

### 3.1. The Procedure of Production Triple-Knockout JNPs

A total of nine surrogate females received 1564 rebuilt embryos via surgical transfer, and three showed signs of early pregnancy. The gRNAs for gene editing are summarized in Table 1, and the schematic process for the creation of triple-knockout JNPs is shown in Figure 1A. The details for the primers of *GGTA1*, *CMAH*, and *B4GALNT2* are listed in Table 2. The sequences of *GGTA1*, *CMAH*, and *B4GALNT2* were obtained from the NCBI database, and deep sequence analysis of the gene target region in the delivered piglets was performed. Exon 7 (*GGTA1*), exon 2 (*CMAH*), and exon 4 (*B4GALNT2*) target sequences were chosen (Figure 1B,C). The production efficiency of triple-knockout (TKO) JNPs is summarized in Table 3 and Table 4. Five TKO JNPs were delivered by two surrogate mothers, whereas one surrogate mother experienced an intermediate abortion (Figure 1D). The success rate of this single transfer was 2.3% and the overall cloning efficiency (total piglets/total transplanted embryos) was 0.5%. The TKO development method was divided into several different strategies. We employed a gRNA method (Figure 2A). A primary cell was cultured from a 16-day-old Jeju native pig. The pSpCas9(BB)-2A-Puro vector and gRNA targeting *GGTA1*, *CMAH*, and *B4GALNT2* treated with BsrI and RFLP were used in the investigation (Figure 2B–E and Appendix A). Finally, the TKO cell line (triple-023 cell line) was constructed (Figure 2F). The matured oocytes were developed in vitro from the immature oocytes (Figure 2G,H). SCNT was executed using expanded COC, giving rise to the blastocyst in vitro (Figure 2I), and the embryo was transferred for the development of healthy offspring (Figure 2J,K).

### 3.2. Expression Patterns of α-Gal, Neu5Gc, and Sd(a) in the Kidneys, Heart, Lung, and Liver of TKO JNPs

The structure and cell morphology of each tissue did not differ between the wild-type and TKO JNPs (Figure 3). The distribution of α-Gal, Neu5GC, and Sd(a) antigens in the tissues of wild-type and TKO JNPs was analyzed using immunofluorescence. As shown in Figure 4 and Figure 5, the intense expression of the three glycans was confirmed in all the tissues of wild-type JNPs, but the intensity of expression was different in each tissue. In the kidney tissue of wild-type JNPs, α-Gal and Neu5Gc were sporadically distributed, whereas Sd(a) showed a partially dense distribution (Figure 4A). In addition, the distribution of the three glycans in the kidney tissue of J04-01 and J04-02 TKO JNPs was significantly less than that of the wild-type TKO, but the expression was not completely eliminated (Figure 4A). In the heart tissue of wild-type JNPs, the three glycans showed increased distribution along the cardiac muscle, whereas their distribution in the TKO JNPs was significantly reduced (Figure 4B). In particular, compared with the cardiac tissue of J04-01 TKO JNP, that of J04-02 TKO JNP showed greater deficiency of α-Gal (Figure 4B). The lung tissue of wild-type JNPs showed basal expression of the three glycans; however, the basal expression of α-Gal was weaker in the lung tissues compared with that in other tissues (Figure 5A). In the lung tissue of the TKO JNPs, α-Gal and Sd(a) were significantly reduced, whereas Neu5Gc was weakly expressed (Figure 5A). The liver tissue of wild-type JNPs showed strong α-Gal and Neu5Gc expression, whereas Sd(a) expression was reduced compared with that in other tissues (Figure 5B). The liver tissue from TKO JNPs showed significantly reduced expression of both Neu5Gc and Sd(a), whereas a trace amount of α-Gal was observed (Figure 5B). Although different for each tissue, the expression of the three glycans (α-Gal, Neu5Gc, and Sd(a)) in the kidneys, heart, lungs, and liver of TKO JNPs was significantly reduced compared with that in the corresponding organs of wild-type JNPs.

### 3.3. Human IgM/IgG Binding Analysis of the Kidneys, Heart, Lungs, and Liver of TKO JNP

Human serum IgG and IgM binding assays for the kidneys, heart, lungs, and liver of wild-type and TKO JNPs were performed (Figure 4 and Figure 5). In the wild-type JNPs, there was no significant difference in IgG and IgM binding in the kidney, heart, and liver tissue; however, IgM in the lung tissue showed stronger binding than IgG. There was no significant difference in IgG and IgM binding in the tissues of the TKO JNPs; the lung tissue exhibited slight IgM binding, whereas the other tissues did not exhibit binding. Although there were differences between tissues, human serum analysis showed that both IgG and IgM binding in the kidneys, heart, lungs, and liver were significantly reduced in the TKO JNPs compared with those in the wild-type JNPs. IgM binding was completely abolished in the kidneys, heart, and liver of the TKO JNPs. However, compared with normal JNPs, TKO JNPs exhibited lower IgM binding in the lungs (Figure 4 and Figure 5). IgG binding was completely abolished in the kidneys of the TKO JNPs. However, IgG binding in the heart, lungs, and liver, while not completely abolished, was significantly inhibited in TKO JNPs compared with that in wild-type JNPs.

## 4. Discussion

Owing to their anatomical and physiological similarities to humans, pigs are being studied as a source of biological materials to curb the shortage of organs for transplantation in humans. However, immune rejection must be eliminated to enable the safe and effective transplantation of pig organs into humans [24,25,26]. A prolonged immune reaction known as acute vascular rejection occurs within a few days to weeks post-transplantation. The reaction involves the infiltration of pre-existing antibodies into the transplanted tissue, complement activation, the pro-inflammatory and pro-coagulator activation of the vascular endothelium, and thrombotic microangiopathy.

Since the production of knockout pigs with the removal of the first immune rejection factor, *GGTA1* was reported [9,24,26], many technological advances have occurred, and the development of the CRISPR system, a powerful gene-editing technology, has made it possible to edit multiple genes in a short period of time [27,28,29]. Recent developments include the production of GT knockout (KO) pigs [10], *GT/B4GALNT2* KO [8], *GTKO/CD46* transgenic pigs [30], *CTKO/CD46/NeuGc* KO pigs [31], *GTKO/CD46/TBM* transgenic pigs [7], *GGTA1/CMAH* KO pigs [32,33], and *GTKO/CD46/CMAH* KO transgenic pigs [31]. When compared with wild-type pigs, pigs with the *GGTA1*/*CMAH*/*B4GALNT2* triple gene deletion showed significantly reduced binding of human IgG/IgM in PBMCs and erythrocytes as well as a steady decline of porcine xenoantigens recognized by human immunoglobulins. The production of TKO pigs with the removal of three genes (*GGTA1*, *CMAH*, and *B4GALNT2*) encoding pig antigens using CRISPR-Cas9 has been reported [34,35]. Using CRISPR-Cas9, we succeeded in producing TKO pigs by the nuclear transplantation of pig fibroblasts in which three immune rejection-inducing antigen genes were removed. Five live offspring were produced from two surrogate mothers, two for immune-rejection-inducing antigen analysis and two for organ transplantation in non-human primates.

In pig-to-human xenotransplantation, porcine xenoantigens such as α-Gal epitopes can activate the binding of human natural IgGs and IgMs, causing immune reactions and leading to hyperacute rejection (HAR). It has been demonstrated that the *GGTA1* knockout successfully reduces HAR [36]. Because the ratio of human IgM binding to fibroblasts in GBC-modified pigs was reduced, we were able to confirm that *GGTA1* attenuates HAR [37]. In this study, we showed that the simultaneous elimination of *GGTA1*, *CMAH*, and *B4GALNT2* inhibited the binding of human IgG and IgM, essentially overcoming HAR [37]. We hypothesized that two additional xenoantigens, Neu5Gc and Sd, together with α-Gal epitopes, should be deleted in multiple genetically manipulated donor pigs for xenotransplantation. It was revealed that TKO JNPs show decreased human IgG and IgM binding in particular tissues. We found that IgG and IgM binding were reduced in the kidneys, heart, liver, and lungs of TKO JNPs. However, they showed weak IgG binding in the heart, lungs, and liver. Compared to the results of Wang et al., this study found marginally different results in the heart and lungs [35]. The expression of all three antigens (α-Gal, Neu5Gc, and Sd(a)) was not observed in the heart, lungs, and liver of TKO JNPs, while poor expression of α-Gal and Neu5G was observed in the kidneys. In contrast, all three antigens were well expressed in the kidneys, heart, lungs, and liver of wild-type pigs. These findings demonstrate that immune rejection can be manipulated by gene editing. The successful use of the proposed gene deletion for xenotransplantation can be confirmed with additional research results, such as non-human primate transplantation. However, the removal of the three genes eliminated or significantly reduced xenograft rejection, suggesting an increase in the production potential of pig organs for xenotransplantation.

## 5. Conclusions

Multiple gene modifications, including human gene knockout or knockin, are required for the functioning of xenotransplanted organs. Our method of producing knockout *GGTA1*/*CMAH*/*B4GALNT2* pigs with mutations in their germ lines but no antigen-related hyperacute rejection is a viable alternative to SCNT and does not require complicated procedures. However, no pigs with multiple gene modifications have been produced using this method. Further improvements aimed at the generation of more multiple-gene modification cascades including the knockout of human genes in porcine zygotes are required to generate genetically modified pigs for pig-to-human xenotransplantation.

## Figures and Tables

**Figure 1 vaccines-10-01503-f001:**
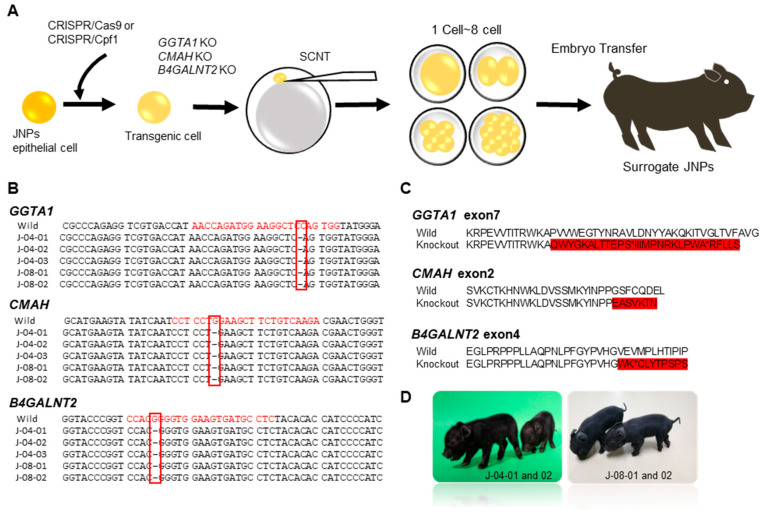
Schematic workflow of the generation of triple-knockout Jeju Native Pigs (JNPs), including gene editing strategies and steps required to obtain the modified JNPs. (**A**) Schematic diagram for triple-knockout development; (**B**) the sequences of *GGTA1*, *CMAH*, and *B4GALNT2*; (**C**) deep sequence analysis of the target region of genes in delivered piglets and details of the target sequences in exon 7 (*GGTA1*), exon 2 (*CMAH*), and exon 4 (*B4GALNT2*); (**D**) knockout piglets produced. SCNT, somatic cell nuclear transfer. (**B**) Red color nucleotide sequences: sgRNA and PAM sequences; (**B**) Red box indicate deleted nucleotide base sequence by RNP complex; (**C**) red highlighted sequences indicate mutated amino acid.

**Figure 2 vaccines-10-01503-f002:**
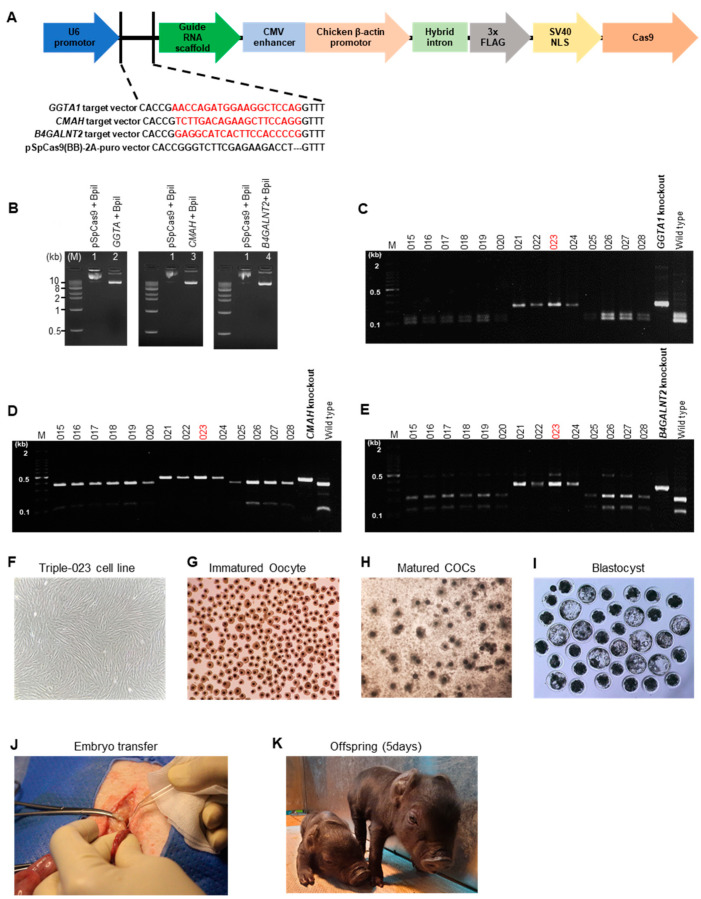
Procedure for triple-knockout development. (**A**) Schematic representation of designing gRNA (vector schematic diagram including Guide RNA); (**B**) preparation of vector expressing Cas9 and gRNA targeting triple gene; M: 1 kb DNA ladder, Lane 1: pSpCas9(BB)-2A-Puro vector treated with BpiI, Lane 2: *GGTA1* target vector treated with BpiI, Lane 3: *CMAH* target vector treated with BpiI, Lane 4: *B4GALNT2* target vector treated with BpiI; (**C**–**E**) RFLP analysis of *GGTA1*, *CMAH*, and *B4GALNT2*, where the PCR product from each single cell was treated with BsrI; (**F**) constructed TKO cell line; (**G**) preparing immature oocyte; (**H**) expanded cumulus cell–oocyte complex at day 2 after in vitro maturation; (**I**) blastocyst was developed at day 7 after SCNT; (**J**,**K**) embryo transfer and development of healthy offspring. (**A**) Red color nucleotide sequences: sgRNA and PAM sequence; (**C**–**E**) 023 indicate selected construct.

**Figure 3 vaccines-10-01503-f003:**
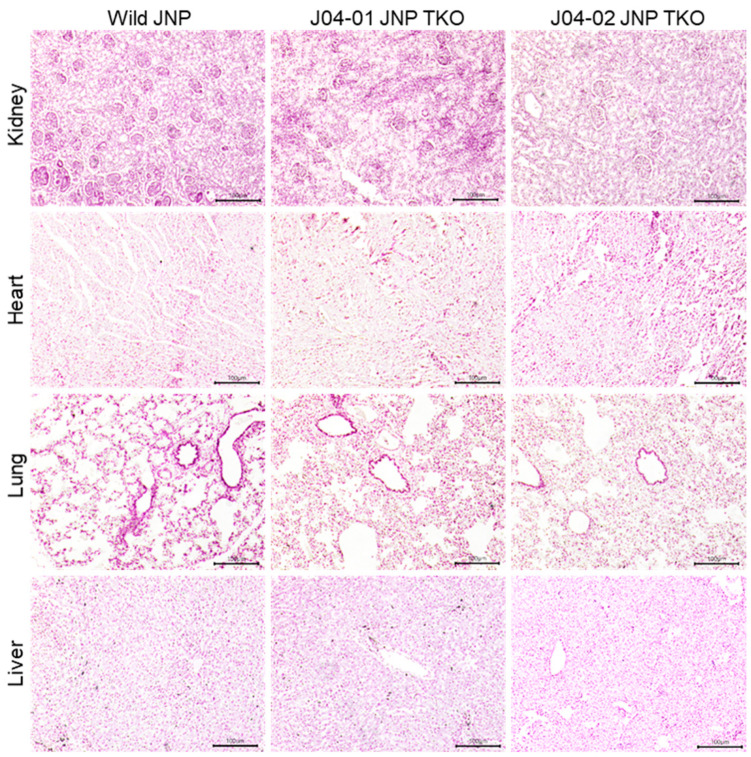
Histology of the primary organs of TKO JNPs. Paraffin-embedded kidney, heart, lung, and liver tissues from wild-type and TKO JNPs were sectioned and hematoxylin and eosin (H&E) staining was performed. Scale bar = 100 μm.

**Figure 4 vaccines-10-01503-f004:**
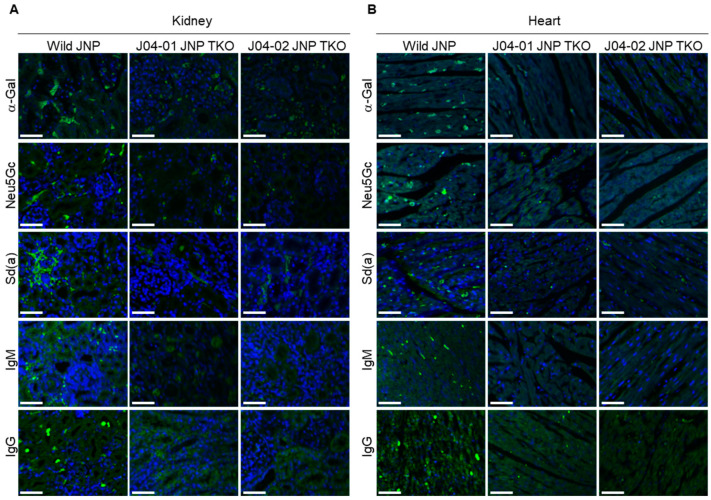
Expressions of α-Gal, Neu5Gc, and Sd(a) along with human IgM and IgG antibody binding assays in the kidney and heart of wild-type and TKO JNPs. Immunofluorescence analysis of α-Gal, Neu5Gc, and Sd(a) was performed using the kidney (**A**) and heart (**B**) of wild-type and TKO JNPs. In addition, human IgM and IgG binding assays were performed using the kidney and heart tissues. Represented images were overlaid with blue (DAPI) for nucleus and green (Alexa Fluor 488) for target proteins (α-Gal, Neu5Gc, and Sd(a)). Scale bar = 30 μm.

**Figure 5 vaccines-10-01503-f005:**
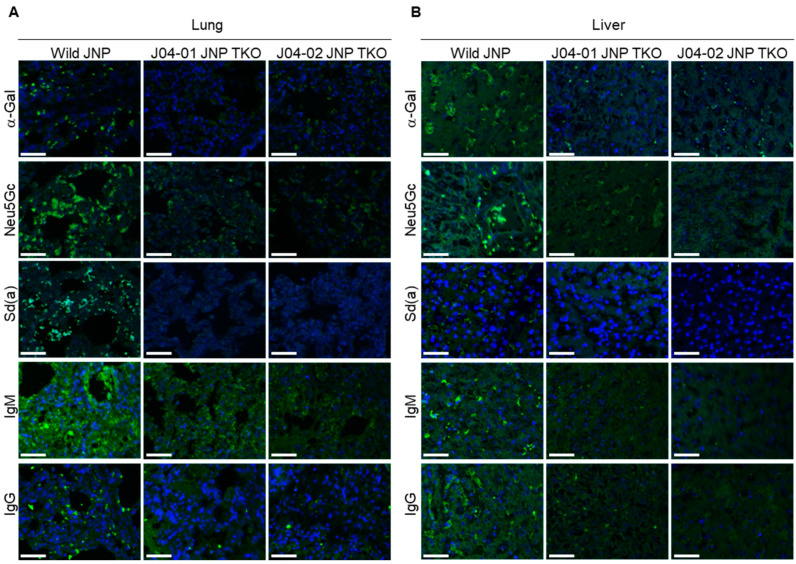
Expressions of α-Gal, Neu5Gc, and Sd(a) along with human IgM and IgG antibody binding assays in the lungs and liver of wild-type and TKO JNPs. Immunofluorescence analysis of α-Gal, Neu5Gc, and Sd(a) was performed using wild-type and TKO JNP lung (**A**) and liver (**B**) tissues. In addition, human IgM and IgG binding assay was performed in the lung and liver tissues. Represented images were overlaid with blue (DAPI) for nucleus and green (Alexa Fluor 488) for target proteins (α-Gal, Neu5Gc, and Sd(a)). Scale bar = 30 μm.

**Table 1 vaccines-10-01503-t001:** List of target single-guide RNAs (sgRNAs).

sgRNA (Name_Target Exon)	Sequences (5’-3’)	On-Target Score	Off-Target Score
*GGTA1*_7	AACCAGATGGAAGGCTCCAGTGG	70.5	35.7
*CMAH*_4	CCT CCTGGAAGCTTCTGTCAAGA	72.1	32.6
*B4GALNT2*_4	GAGGCATCACTTCCACCCCGTGG	74.3	40.3

**Table 2 vaccines-10-01503-t002:** List of primers used for PCR and sequencing.

Name	Sequence (5’-3’)	Annealing Temp. (°C)	Product Size (bp)
*GGTA1*_7F	GGATTCAAGGCCAGTCACCA	59	241
*GGTA1*_7R	CCTTCCGACAGCAAAAACCG	59.1	
*GGTA1*_7F_Se	CCAGCTGACTGGGGCTAAAA	60.0	408
*GGTA1*_7R_Se	AAAATGGCCCTGTGACACCA	60.1	
*CMAH*_4F	GCTCTGCTGATCTCTAACACG	58.5	520
*CMAH*_4R	GTTGACAAGAGGGACCCCAA	59.5	
*CMAH*_4F_2	AGTCAGGGAAACACGAAGAGTC	59.9	458
*CMAH*_4R_2	ACAAGAGGGACCCCAATGAC	59.3	
*B4GALNT2*_4F	CAGGGACAGGTATCAAGGCA	59.1	324
*B4GALNT2*_4R	CAGTGGTGGAAACAGTGAGA	57.4	
*B4GALNT2*_4F_Se	AAGAACGAACCAGTGGGAGC	60.2	467
*B4GALNT2*_4R_Se	CCATGTCCAGCTTCACGGAT	60.1	

**Table 3 vaccines-10-01503-t003:** Production of triple-knockout piglets.

No. of Embryos Transferred	No. of Recipients (%)		No. of Cloned Piglets	Live Birth	No. of Knockout Piglets	Weight * (g)
	Day 30	Delivered				
1564	3/9 (33.3)	2/9 (22.2)	5	5	5	627.8 ± 100.2

* means ± SE.

**Table 4 vaccines-10-01503-t004:** Full-term development of triple-knockout piglets.

ET No.	No. of Transferred Embryos	Day 30 Pregnancy Status *	No. of Piglets Born (Knockout)	Specificity
J-01	204	+		Abortion
J-02	205	−		
J-03	166	−		
J-04	178	+	3 (3)	3 live births
J-05	153	−		
J-06	149	−		
J-07	161	−		
J-08	172	+	2 (2)	2 live births
J-09	176	−		

* +, Pregnant; −, not pregnant.

## Data Availability

The gene sequence data for GGTA1, CMAH, and *B4GALNT2* that were used in the study were obtained from the NCBI database (accession numbers NC_010443, NC_010449, and NC_010454, respectively). Further, the data presented in this study are available on request from the corresponding author.

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
