# Peer review of "An Efficacious Transgenic Strategy for Triple Knockout of Xeno-Reactive Antigen Genes GGTA1, CMAH, and B4GALNT2 from Jeju Native Pigs"

_vaccines, 2022, doi:10.3390/vaccines10091503_

Round 1

Reviewer 1 Report

In this study,target GGTA1CMAH, and B4GALNT2 were knockout using CRISPR-CAS9 in Jeju Native Pigs.They found that IgG and IgM binding were reduced in the kidneys, heart, liver, and lungs of TKO JNPs. However, they showed weak IgG binding in the heart, lungs, and liver.This paper showed that CRISPR-CAS9 maybe an  effective method to knockout swine endogenous genes. The authors should design more effective gRNAs to make target GGTA1, CMAH, and B4GALNT2 knockout clearly.

Author Response

Reviewer #1:

In this study, target GGTA1, CMAH, and B4GALNT2 were knockout using CRISPR-CAS9 in Jeju Native Pigs. They found that IgG and IgM binding were reduced in the kidneys, heart, liver, and lungs of TKO JNPs. However, they showed weak IgG binding in the heart, lungs, and liver. This paper showed that CRISPR-CAS9 maybe an effective method to knockout swine endogenous genes. The authors should design more effective gRNAs to make target GGTA1, CMAH, and B4GALNT2 knockout clearly.

[Response]

We concur with the remarks made by reviewer #1. For more successful target gene knockout, we have already developed 25 SCNT cell lines which are designed for other target exons of GGTA1, CMAH, and B4GALNT2. Consider the fact that a generation of JNPs with 6 genes knocked out is our ultimate goal. As a result, we assumed that the effectiveness of the gene knockout used in this study was sufficient to produce six gene knockout JNPs.

Reviewer 2 Report

The submitted work is of a very good standard and I have only minor comments.

In parts material and methodology I recommend providing more information about experimental animals. The article states that piglets were euthanized at the age of 16 days. This means that sows and piglets were present or the piglets were already weaned at the age of 16 days? Please state how many sows were used and what feed was given to the sows. What feed was given to piglets? Please state how many piglets were used in total and from how many sows they came from.

Author Response

Reviewer #2:

The submitted work is of a very good standard and I have only minor comments.

In parts material and methodology I recommend providing more information about experimental animals. The article states that piglets were euthanized at the age of 16 days. This means that sows and piglets were present or the piglets were already weaned at the age of 16 days? Please state how many sows were used and what feed was given to the sows. What feed was given to piglets? Please state how many piglets were used in total and from how many sows they came from.

[Response]

We are thankful to the reviewer for supporting us in further improvement of this manuscript. We used nine sows for generation TKO JNPs, and two sows were successfully delivered. Finally, we got five piglets. Please see Table 4. The piglets were suckled with a mother sow, and we used standard sow feed.

Reviewer 3 Report

In this study, the authors targeted the porcine genes GGTA1, CMAH, and B4GALNT2 using the CRISPR-Cas9 system and introduced α‐Gal, Neu5Gc, and Sd deficiency by somatic cell nuclear transfer (SCNT) to reduce antibody-mediated xenograft rejection after xeno-transplantation. They found that reducing the expression of swine xenogeneic antigens identifiable by human immunoglobulins can lessen the immunological rejection against xenotransplantation. It was an interesting and well-organized study. One concern was listed here as below.

1) It was indicated that the structure and cell morphology of each tissue did not differ between the wild-type and TKO JNPs (Figure 3). However, 1) for example, the numbers of glomerulus were less in the TKO JNPs groups than in wild-type group; Massive inflammation cells infiltration seem to be present in the TKO JNPs groups. 2) All of these HE images appeared to be indistinct. So, the authors need to ask for the experienced pathologists to check these figures and, 1) More clear images should be provided; 2) Larger magnification of images should also been added.  

Author Response

Reviewer #3:

In this study, the authors targeted the porcine genes GGTA1, CMAH, and B4GALNT2 using the CRISPR-Cas9 system and introduced α‐Gal, Neu5Gc, and Sd deficiency by somatic cell nuclear transfer (SCNT) to reduce antibody-mediated xenograft rejection after xeno-transplantation. They found that reducing the expression of swine xenogeneic antigens identifiable by human immunoglobulins can lessen the immunological rejection against xenotransplantation. It was an interesting and well-organized study. One concern was listed here as below.

Comment-1

1) It was indicated that the structure and cell morphology of each tissue did not differ between the wild-type and TKO JNPs (Figure 3). However, 1) for example, the numbers of glomerulus were less in the TKO JNPs groups than in wild-type group; Massive inflammation cells infiltration seem to be present in the TKO JNPs groups.

[Response]

Thank you for reviewer #3’s critical comment. In this study, we focused on the knockout efficiency of three key antigens, e.g., α-Gal, Neu5Gc, and Sd, and analyzed human IgG/IgM binding activity. We are investigating the physiological and pathological functions of the main organs.

Comment-2

2) All of these HE images appeared to be indistinct. So, the authors need to ask the experienced pathologists to check these figures and, 1) More clear images should be provided; 2) Larger magnification of images should also been added. 

[Response]

Thank you for reviewer #3’s helpful comment. As we mentioned in reviewer #3’s comment-1, we focused on the knockout efficiency of three key antigens and tested human IgG/IgM binding activity. We are investigating the physiological and pathological functions of the main organs with an expert veterinarian. Following the reviewer’s comment, we have replaced H&E images with clear ones. We do not have a suitable larger magnification of H&E images.

Round 2

Reviewer 1 Report

That's OK !